# Position: AI Lock-In Is in Progress, and We Must Be Prepared

**Jaeho Kim** [* 1]  **Seokhyun Lee** [* 1]  **Jieun Lee** [* 2]  **Changhee Lee** [1]

## Abstract

AI safety research has mainly focused on two areas: technical alignment (ensuring AI systems produce human-aligned outputs) and the regulation of generative AI's societal impacts (including unemployment risk and labor market disruption). However, an equally important dimension remains underexplored: the risk inherent in dependence on AI systems themselves. In this position paper, we argue that AI safety research should address *AI Lock-In*, the phenomenon whereby excessive reliance on AI systems leads to human deskilling, diminishes human capacity for independent functioning, and creates systemic vulnerabilities when AI systems become unavailable or compromised. We highlight that AI Lock-In is a systemic threat that is already emerging at individual, societal, and national levels, one that could be dramatically amplified by AI service disruptions or geopolitical conflicts. Drawing on detailed scenarios, we investigate how AI Lock-In emerges and escalates across multiple levels, ranging from individual skill atrophy to national-scale infrastructure failures. To address this, we provide guidance on how such risks can be mitigated and prepared for at each level. We contend that proactively addressing AI Lock-In before such dependencies become entrenched, or even irreversible, is essential for preserving individual autonomy and national security.

## 1. Introduction

The rapid adoption of artificial intelligence (AI) and its downstream applications by the general public has been unprecedented in technological history. According to the Stanford HAI Index 2025 report (Maslej et al., 2025), ap-

> **AI Lock-In**
>
> The **AI Lock-In** refers to a state in which individuals and society's reliance on AI becomes so deeply rooted – cognitively, culturally, economically, and infrastructurally – that the costs of switching to alternatives (*i.e.,* human skills) grow increasingly prohibitive over time, eventually making reversion to a pre-AI state nearly impossible. When AI Lock-In occurs, society is forced to accept the associated risks, regardless of emerging concerns or predictable consequences.

proximately two-thirds of the global population now expect AI-powered products and services to significantly affect their daily lives within the next three to five years.

With such explosive growth, AI safety research has focused on addressing the risks accompanying this technological advancement: ensuring AI outputs are safe and aligned with humanity's values (Weidinger et al., 2021; Ouyang et al., 2022; Kulveit et al., 2025), developing AI systems capable of refusing harmful instructions (Bai et al., 2022), and measuring AI's impact on society and mitigating the associated risks (Hazra et al., 2025; Appel, 2025). While these research directions provide useful guidelines on how AI should be trained and how its societal impacts should be managed, *they do not address the threat that emerges from our growing dependence on AI systems.* As such, **this position paper argues that AI safety research and regulation must address AI Lock-In**. Over-reliance on AI systems by individuals and society leads to AI Lock-In: the gradual deskilling of essential cognitive tasks at the individual level, the erosion of human capabilities at the societal level, and a future shaped by risks we cannot yet fully predict or are not prepared for.

As AI Lock-In progresses, individuals and society will grow increasingly dependent on AI, exposing vulnerabilities to sudden accidental failures, cyber attacks, and geopolitical coercion. We believe that AI Lock-In is not a speculative future risk but an ongoing reality that is already manifesting across various domains, evidenced by observable skill atrophy in AI-dependent professions and escalating strategic dependencies that drive a new geopolitical AI arms race.

Much like a snowball rolling down a slope, this risk com-

---

*Equal contribution [1]Department of Artificial Intelligence, Korea University, Seoul, South Korea [2]KRAFTON, Inc., Seoul, South Korea . Correspondence to: Changhee Lee <changheelee@korea.ac.kr>.

*Proceedings of the 43rd International Conference on Machine Learning*, Seoul, South Korea. PMLR 306, 2026. Copyright 2026 by the author(s).

**Table 1.** We compared the Lock-In progression stage-by-stage between biological specialization observed in koalas and AI adoption. We posit that the specialization trajectory of the koala's eucalyptus dependency closely mirrors how AI Lock-In progresses in current society.

| Stage | Biological Specialization (Koala) | AI Adoption |
|---|---|---|
| Introduction | Generalist ancestors discover eucalyptus with few competitors | AI introduced as productivity-enhancing tool |
| Perception | One beneficial food option among many; alternatives remain viable | Useful but not indispensable; imperfect outputs |
| Expansion | Specialized enzymes and digestive system evolve for efficiency | Rapid improvements across domains |
| Behavioral Shift | Eucalyptus preference becomes genetically fixed | AI use becomes default behavior |
| Alternatives Erode | Digestive traits for other foods atrophy from disuse | Human skills atrophy from disuse |
| Lock-In | Koalas can't survive without eucalyptus; now face extinction | Society is dependent despite emerging risks |

pounds over time: the longer we delay our action, the more deeply society becomes AI-Locked in. In such a state, the cost of reverting may become so prohibitive that society is left with no choice but to endure discomforts and risks, potentially exposing humanity to dangers that may threaten the long-term sustainability of both individuals and societies. Through this paper, we posit that we can alleviate, or even prevent AI Lock-In at the individual, organizational, and national levels through the following measures:

- **Individuals.** We advocate for reconceptualizing AI literacy as "literacy for the AI era," which includes two complementary dimensions: *AI-engaged literacy*, the capacity to live *with* AI, and *AI-independent literacy*, the capacity to live *without* AI. Practicing both types of literacy is essential as it enables users to engage with AI critically while preserving human capability.

- **Organizations.** We urge organizations to adopt a *bet-hedging* strategy in AI incorporation: intentionally avoiding maximum short-term efficiency by keeping a proportion of human workers in roles that AI could replace, and continuing to hire and train junior employees. While this approach may lose immediate productivity gains, it preserves the human tacit knowledge and strengthens organizations' resilience in AI Lock-In.

- **Nations.** We urge nations proactively incorporating AI services across all sectors of society to prioritize *AI resilience*: (i) mandating that critical infrastructure maintain operational capability without AI, (ii) incentivizing organizations to adopt bet-hedging strategy in AI use, and (iii) enforcing periodic "AI-free drills", similar to fire drills, where individuals and organizations perform tasks without AI assistance, preparing for sudden disconnection to AI services.

Our position paper is outlined as follows. We first introduce the Lock-In theory in Section 2, using biological example to show how early commitments can become impossible to reverse, potentially leading to severe risk. We then describe how AI Lock-In is quietly but deeply taking root across individual, organizational, and national levels in Section 3. Subsequently, we illustrate how this risk can materialize and

be felt at all levels in Section 4. Consequently, Section 5 discusses concrete measures that can be taken to safeguard from AI Lock-In across all subjects. We examine alternative views to our position in Section 6. We provide our discussion points and limitations in Section 7, and we conclude our position in Section 8.

## 2. The Lock-In Theory

Lock-In (Arthur, 1989) is an economic term describing the tendency to become dependent on a technology that initially appears as one of many options, but over time expands its position and influence until it becomes the only choice. At the early stage of Lock-In, switching to alternatives is easy, but would seem rather irrational and unattractive, given the technology's clear and immediate advantages (Zauberman, 2003). However, over time, other options disappear entirely – whether by design or by circumstance. Eventually, users find themselves *Locked-in* to the technology. Once users are Locked-in, they are forced to rely on the specific technology, giving rise to new and often unforeseen risks.

Interestingly, a parallel to Lock-In can be found in evolutionary biology through *biological specialization*, the process by which organisms adapt structurally to exploit specific skills or resources. The koala provides a representative example. Research shows that ancestral koalas were once capable of consuming a diverse range of foliage (Louys et al., 2009). However, koalas discovered that eucalyptus leaves, which contain toxins that make them inedible for most other animals, offered an abundant resource with few competitors. Initially, eucalyptus was simply one beneficial food option among many alternatives. Over time, koalas gradually evolved specialized enzymes and an elongated cecum capable of digesting these leaves with greater efficiency. As this preference became genetically fixed, their digestive traits for processing other foods atrophied.

Unfortunately, koala's specialization, driven solely by efficiency, has now brought the species to the brink of extinction (Lunney et al., 2014). Today, as climate change and habitat destruction lead to decreased eucalyptus forests

across Australia, koalas face extinction precisely because they cannot revert to alternatives. The adaptation that once ensured their survival has become their vulnerability.

The example of the koala showcases the danger of Lock-In when alternatives have all but disappeared. Notably, we observe a strikingly similar pattern between the koala's biological specialization to eucalyptus dietary and the AI adoption trajectory in today's society (compared in Table 1). In the introductory stage of AI Lock-In, AI is presented as a productivity-enhancing tool (Achiam et al., 2023). Users perceive AI as a useful tool that can help them write, draw, code, and even complete tasks that were difficult for humans to complete (Phan et al., 2025). Ironically, AI's tendency to make trivial mistakes (Cheng & Yu, 2023), hallucinate (Laban et al., 2025), and to produce biased judgments (Cheung et al., 2025) has delayed full AI Lock-In, making it useful but not yet indispensable. However, over the past few years, we have observed significant improvements in AI model performance, with major benchmarks rapidly becoming obsolete (Liu et al., 2024; Maslej et al., 2025), and new AI services being launched with increasing frequency (Singla et al., 2024). Such advancement will potentially make the use of AI as a default behavior for our everyday tasks, leading us to a gradual behavioral shift.

Society now stands at a critical juncture. Currently, the society remains a *skilled society*, where individuals still possess the expertise and cognitive skills that are essential to performing everyday tasks. However, individuals and organizations are actively adopting, or even institutionalizing (Bansal, 2026), AI to replace tasks that were previously performed by humans, where nations are actively encouraging such offloading to improve productivity. At this stage, AI use becomes the default behavior, and human skills begin to atrophy through disuse. We characterize this as the early stage of AI Lock-In, in which society drifts toward a deskilled future. Importantly, this early stage represents a narrow window for intervention. Society remains skilled for now, but as reliance on AI deepens, the capacity to revert will steadily erode. Once this transition is complete, reversing AI Lock-In may become prohibitively difficult. This is why we call for immediate actions: **AI safety research and regulations must prevent AI Lock-In**.

## 3. The AI Lock-In is in Progress

This section outlines the type of AI dependency we are observing in individual, organizational, and societal levels.

### 3.1. Individual-Level Dependency: Cognitive Offloading Leads to Deskilling

Over the past few years, AI has greatly enhanced productivity for many individuals. According to an economic re-

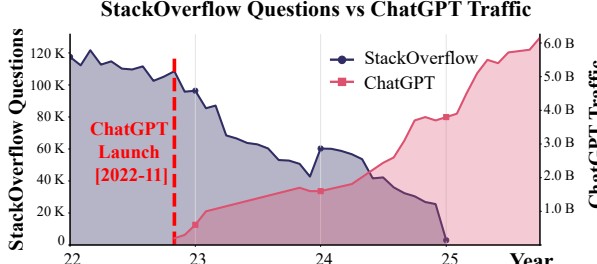

**Figure 1. Number of StackOverflow Posts.** StackOverflow *was* a popular website where users shared technical knowledge and sought help with coding issues. However, since the advent of LLM services (*e.g.,* ChatGPT) in 2022, the number of questions posted has plunged below the year the platform launched (DemandSage, 2026; Smith, 2024). A direct causal link is hard to define, but it is notable that such a decline coincided with the widespread adoption of LLM-based coding assistants (Zhong & Wang, 2024).

search conducted by Anthropic (Tamkin & McCrory, 2025), it was found that Claude, Anthropic's LLM service, has sped up individual tasks completion time by approximately 80%. While the magnitude of productivity gains are uneven across task types, quantitative results clearly indicate large productivity improvements that can be expected at the individual level with AI use. This, in turn, provides strong motivation for users to rely on AI to complete tasks that were previously performed based on human skills (Zhang et al., 2024).

However, this productivity gain comes at a cost. As users increasingly rely on AI for task completion, it gives rise to *cognitive offloading* (Risko & Gilbert, 2016; Kulveit et al., 2025), whereby individuals reduce cognitive effort by externalizing information processing. The concern is that such offloading to AI now spans a broad range of cognitive functions: memory, attention, reasoning, problem-solving, and even basic writing. For instance, the number of coding-related questions posted on StackOverflow (see Figure 1) has declined by 60% since the release of ChatGPT. While multiple factors may have contributed to this decline, it coincides with a broader shift toward using LLMs for coding assistance (Burtch et al., 2024).

More importantly, continuous cognitive offloading often leads to *deskilling*, the atrophy of human skills as they are no longer practiced in domains where humans were once proficient (Carr, 2014; Ferdman, 2025a;b). Evidence of such deskilling is already emerging. For instance, in the field of medicine, the adenoma detection rate by doctors in non-AI assisted procedures declined after being exposed to AI-assisted colonoscopy just after three months (Budzyń et al., 2025). In academics, students exposed to LLM for tutoring initially had better math performance, compared to students who were not exposed to, but underperformed when such access was removed (Bastani et al., 2025). Both examples indicate that the indiscreet use of AI for productiv-

ity can atrophy human skills and gradually lead to stronger AI dependency. This gradual decay of human capability is concerning, as autonomous thought and skilled action are central to sustained human competence and independence.

Crucially, AI dependency does not remain an individual problem. As AI is adopted across individuals and institutions, dependency at the individual level aggregates into collective vulnerabilities, giving rise to more structural forms of AI Lock-In at organizational and national levels. This transition from individual to systemic dependency is accelerated when organizations and nations actively incentivize or mandate AI usage as a performance target (Maslej et al., 2025; Bansal, 2026), removing individual agency over the offloading decision and enforcing dependency at scale. We elaborate on this in the following section.

### 3.2. Systemic Dependency: Organizational and National-Level AI Lock-In

In this subsection, we center our analysis on the dependency that organizations gradually develop on AI. While important, we intentionally defer discussions on the implications of AI on the future of work (Hazra et al., 2025), income inequality (Goyal & Aneja, 2020), and related ethical consequences (Gratch & Fast, 2022), as our work focuses on the dependency risk in AI itself.

**Organizational Level.** As AI become increasingly capable of understanding human instructions (Wiesinger et al., 2024), organizations are rapidly embedding AI into their workflows. According to a UK financial sector survey in 2024, 75% of financial firms are already using AI, with an additional 10% planning to adopt AI within the next three years (Gharbawi et al., 2024). Similarly, in the healthcare industry, 85% of 150 U.S. healthcare leaders reported that they were either exploring or had already adopted AI capabilities into their workflow (Pardo Martin & Lamb, 2025).

Yet, such rapid adoption has not been without adverse consequences. In the short term, organizations have faced operational failures and legal liabilities: Air Canada was ordered to pay compensation after its AI chatbot provided incorrect information due to hallucination (Cecco, 2024); New York City's official chatbot was criticized for providing inaccurate policy guidance that, in some cases, recommended potentially illegal actions (Lecher, 2024). Of greater concern, the aggressive substitution of tasks with AI has coincided with a marked decline in junior-level hiring, with unemployment rate for new college graduates rising to 30% since September 2022 (Federal Reserve Bank of New York, 2025). In contrast, demand for managerial roles remains robust or even increasing (Alekseeva et al., 2024). A Stanford Digital Economy Lab study offers an explanation: AI disproportionately substitutes for codified knowledge acquired through formal education and digital records, while it struggles to replace tacit knowledge: the experiential "tips and tricks" accumulated over time. Since junior workers tend to supply relatively more codified knowledge, they face greater displacement (Brynjolfsson et al., 2025).

As such, current organizational AI lock-in can be conceptualized as occurring via two interrelated pathways: (i) an increasing reliance on AI-embedded workflows and, (ii) a concurrent erosion of human capital as entry-level positions are progressively substituted by AI. These dependency pathways reinforce one another and create a reiterative loop, which in turn fosters organizations' long-term reliance on AI and diminishes their capacity to operate without it. As organizations adopt AI across a broader range of tasks, a decline in entry-level hiring may intensify, leaving fewer opportunities for junior employees to receive formal training and to acquire seniors' tacit knowledge (Ide, 2025). This dynamic may create gaps in the workforce pipeline, as fewer workers progress into senior roles; consequently, even if an organization later seeks to operate without AI, it may lack sufficiently experienced staff to do so, potentially making this AI dependency more deeply rooted.

**National Level.** At the national level, countries are actively encouraging the deployment of AI services to boost productivity and economic growth (Maslej et al., 2025). For instance, the UK has launched the "AI Opportunities Action Plan" in January 2025 to increase AI adoption across the country and scale AI products into both public and private sectors. Meanwhile, the United States has partnered with OpenAI to provide ChatGPT Enterprise to federal agencies and national laboratories at minimal cost, supporting administrative operations and frontier research. China is also actively investing in national AI capabilities by encouraging the adoption of its homegrown LLM service, DeepSeek, across courts, hospitals, and city governments for legal drafting, treatment planning, and public administration, respectively (Deng et al., 2025). These national-level policies and subsidies reflect a global AI race, driven by competitive anxiety over the belief that AI adoption across both private and public sectors is essential for national advancement.

Unfortunately, these nation-led AI policies and the active encouragement to deploy and use AI throughout the entire society elevate cognitive offloading from the individual level to the national level, accelerating deskilling not just among individuals but across the entire workforces and public institutions. What is worrisome is that AI adoption may *initially* appear as a matter of choice. Organizations and nations will begin with pilot tests, applying AI to a limited set of tasks, before gradually expanding its use across broader operations. However, beyond a tipping point, this optional adoption will transition into a structural dependency where functioning without AI becomes virtually unimaginable, finding ourselves in a state of AI Lock-In.

When such dependency in AI is deeply rooted, we ask: what happens when AI suddenly stops functioning, whether due to unforeseen accidents, coordinated cyberattacks, or deliberate geopolitical coercion? In such a disastrous scenario, everything will come to a halt. Office workers who relied on AI for their daily tasks see their productivity plummet; developers unable to write working code are forced to stop working. AI-dependent traffic lights malfunction and autonomous vehicles come to a standstill in the middle of the road, paralyzing the entire transportation network. Logistics systems collapse with no human workers available to sort and deliver packages. Government offices, waiting for AI to be restored, decide to close doors as simple administrative tasks once handled by AI pile up unprocessed. Some may dismiss this scenario as science fiction, arguing that it is far too unlikely that AI can suddenly stop functioning all at once. In the next section, we demonstrate that such a collapse is highly possible, evidenced by events that happened before. Furthermore, we argue that the emergence of AI Lock-In renders these scenarios increasingly realistic.

## 4. Threat Scenario

This section outlines three scenarios that can lead to AI disruption: (i) accidental failures across systems, (ii) cyber attacks, and (iii) geopolitical coercion. We believe that such disruption will become more frequent and likely as AI Lock-In deepens. We emphasize that the real danger is not the AI failure itself, but human's diminished capacity to respond to such threat scenarios, due to AI dependency.

### 4.1. Accidental Failures Aren't Failures: Normal Accident Theory

Modern AI development is characterized by increasingly complex dependencies on other AI and software services: from multi-agent systems where one agent's (*i.e.,* an LLM's) actions depend on other agents' decisions (Guo et al., 2024), to inference and software update pipelines that embed multiple services from different providers. This interdependency creates precisely the conditions that Normal Accident Theory (Perrow, 2011) warns against. According to the theory, minor errors in a system can be considered statistically normal, and such errors do not necessarily break down the system. However, when multiple systems are tightly coupled, such trivial errors can accumulate to cause catastrophic accidents. While the probability of failure in any single component may be negligible, when numerous components operate together, these small probabilities compound. As such, what emerges is not a question of *if* but *when*.

The "Left-Pad" incident in the JavaScript ecosystem in 2016 illustrates such a case (Abdalkareem et al., 2020). The Left-Pad was part of an open-source package which enabled padded strings with spaces, consisting of only 11 lines of code. However, following the dispute between its developer and npm (the package manager for JavaScript), the developer removed the package from the ecosystem. Surprisingly, thousands of projects and services were disrupted by its removal, including those from major companies such as Facebook, Netflix, and PayPal, leaving millions of users affected for several hours.

The message from this incident is clear. As AI systems become interdependent and humans offload more tasks to AI, the conditions for Normal Accidents become not exceptional but routine. The AI Lock-In risk, in other words, does not merely increase the chance of AI failure; it *normalizes* it. Paradoxically, the AI ecosystem built on numerous open-source libraries (*e.g.,* PyTorch, TensorFlow, Hugging Face) becomes a source of vulnerability when AI Lock-In occurs. To be clear, we are not arguing against open-source development, but rather stressing the need for societal vigilance: we must recognize that we are slowly drifting towards AI Lock-In, and now is precisely the time when society still retains the capacity to choose a different path.

### 4.2. Cyber Attacks: AI as Both Weapon and Target

AI disruption can be amplified, or deliberately engineered through cyber attacks. While AI is contributing to advances in cybersecurity (Sarker et al., 2021), the same capabilities are being exploited by adversaries (Kaloudi & Li, 2020; Lin et al., 2025). According to CrowdStrike's 2025 Global Threat Report (CrowdStrike, 2025), the average time for an attacker to move within a secured network dropped from 62 minutes to 48 minutes compared to year 2023, with generative AI enabling more personalized and targeted attacks. The company also projects that attacks on AI models and AI-based infrastructure will surge significantly over the next two to three years, including attempts to corrupt model parameters or training data to induce malfunctions.

Beyond using AI as a weapon, hackers are exploiting the vulnerabilities in the AI software supply chain (Liu et al., 2026). One recent example was the attack on LiteLLM, a popular Python open-source package used for providing a unified abstract wrapper over different LLM services. Here, adversaries took over the maintainer's account and released packages containing malicious code, leading to leakages of credentials, API, and SSH keys from users who either directly used LiteLLM or relied on external libraries that depend on it. More alarmingly, any package listing LiteLLM as a dependency may automatically pull in the compromised version upon update, propagating the attack across a vast number of downstream AI services and systems. While the attack on LiteLLM was targeted at credential theft, future attacks of this type could be designed to be far more sophisticated and to inflict far greater disruption.

In a society heavily dependent on AI, where humans have

delegated many tasks to AI systems, such attacks against AI infrastructure and supply chains can serve as a means of widespread societal disruption. Unlike accidental failures that may be localized, coordinated cyber attacks backed by groups of hackers, organizations, or even nations (Klingner, 2021), can simultaneously compromise multiple AI systems across sectors, amplifying the damage exponentially.

### 4.3. Geopolitical Coercion

The current global AI ecosystem is heavily reliant on a small number of cloud (*e.g.,* AWS, Google Cloud), GPU (*e.g.,* NVIDIA, CUDA), and API providers (*e.g.,* OpenAI, Google, Anthropic, DeepSeek). This concentration creates vendor Lock-In (Opara-Martins et al., 2016), potentially leading to these services being compromised or leveraged in geopolitical conflicts. Just how United States has imposed semiconductor export regulations on China to limit its technological advancement (Mark & Roberts, 2023), similar restrictions could be used on AI services and infrastructure. For nations that are heavily AI-Locked-In, such disruption to AI services could ripple across every layer of society, from individual healthcare to governmental public administration, leaving entire societies unable to operate.

It is important to note that the problem is not primarily about dependency on foreign AI providers, but about the risks of AI dependency itself. While several nations are investing in domestic cloud infrastructures, chips, and LLM services as part of Sovereign AI initiatives to reduce foreign AI dependency (Maslej et al., 2025), these efforts do not address the fundamental nature of AI Lock-In risk itself. A nation that has achieved complete AI sovereignty but remains deeply AI Locked-In is still vulnerable to the threats mentioned. To reiterate, AI Lock-In risk lies not in *who* provides the AI, but in *how deeply and indiscriminately* a society has integrated AI into its core functions and lost the capacity to operate without it.

The scenarios we have listed above, such as accidental failures, coordinated cyber attacks, and geopolitical coercion, are a highly probable future if AI Lock-In deepens. Yet, as we argued in Section 2, the current society remains a skilled society and is capable of charting a different course. However, with the current pace of AI adoption, the window for intervention is narrowing and actions must be taken.

## 5. Call to Action: Building Resilience

This section outlines practical measures that can be taken at each subject level to mitigate the AI Lock-In.

### 5.1. AI Literacy Education for Individuals

According to the EU's AI Literacy (EU AILit) framework for primary and secondary education, AI literacy is the *tech-nical knowledge, skill, and attitude required to thrive in a world influenced by AI, enabling users to use AI while critically evaluating its benefits and risks* (OECD, 2025). While this definition of AI literacy provides valuable guidance in cases where AI is available, it does not fully address scenarios where AI becomes unavailable. We argue that AI literacy should not be confined to "literacy *about* AI," but more broadly as "literacy *for* the AI era", which necessarily includes both the capacity to thrive *with* AI and the capacity to remain functional *without* it.

Building on this argument, we expand the scope of AI literacy and subdivide it into two dimensions: (i) **AI-engaged literacy** refers to the ability to use and engage *with* AI critically – this has been the primary focus of conventional AI literacy education, and (ii) **AI-independent literacy** is the ability to retain and function *without* AI – the ability to perform tasks independently, even with reduced efficiency or convenience. We highlight that both forms of literacy are essential and complementary. However, practicing one does not naturally develop to the other, nor does the reverse hold. Both must be emphasized equally, and learned in parallel.

**AI-Engaged Literacy.** Proficiency in this dimension means individuals have the capacity to use AI and critically evaluate its output, enabling them to effectively engage *with* AI. Fortunately, AI-engaged literacy education can use existing AI Literacy resources (*e.g.,* EU AILit), and even adapt from other related literacy theories. For instance, drawing on the perspective that traditional media's function is to transmit and produce knowledge (Potter, 2018), AI can be understood as a new form of media: generating, editing, and refining information for users at extreme speed and efficiency. As such, the principles used in teaching media literacy, critically assessing the source of information, understanding how such information was created, can be similarly applied and adapted for AI-engaged literacy education (OECD, 2025). For instance, we show that the Center for Media Literacy's Five Core Concepts, questions that are used to train students' media literacy (Jolls, 2014), can be easily reformulated to such context (shown in Table 2). Through cultivating this literacy, individuals can strengthen their critical judgment skills against AI, while also developing awareness of the risks posed by AI Lock-In.

**AI-Independent Literacy.** Developing AI-independent literacy enables users to function autonomously *without* AI, accepting modest trade-offs in efficiency. Such practice includes the deliberate choice to perform tasks manually that could otherwise be completed with AI, such as writing without AI assistance, reading without AI-based summary, or even tracking one's ability to function without AI for a day. The objective of this practice is not to reject AI entirely, but to constantly practice and maintain the underlying cognitive skills that are being offloaded to AI. Just as physical

**Table 2. Adaptation of the Center for Media Literacy's (CML) Five Core Concepts to AI Literacy.** These questions have long been used to train students in media literacy (Jolls, 2014), so that students can assess and critically evaluate media messages. We advocate that these questions can be adapted for AI literacy education, enabling users to critically engage with AI and its output.

| # | Core Concept | Key Question | Adaptation to AI Literacy |
|---|---|---|---|
| 1 | Authorship | Who created this message? | AI responses are not objective truths but probabilistic constructions shaped by data and algorithms. |
| 2 | Format | What creative techniques are used to attract my attention? | Analyze how AI's natural tone and authoritative voice instill an "illusion of competence" in users. |
| 3 | Audience | How might different people understand this message differently? | Recognize that AI may provide biased information tailored to individuals through filter bubbles and echo chambers. |
| 4 | Content | What perspectives are represented in or omitted from this message? | Critically evaluate cultural biases and ethical standards embedded in AI training data. |
| 5 | Purpose | Why is this message being sent? | Understand how AI companies design systems to maximize user engagement for commercial purposes. |

muscles atrophy without use, cognitive skills such as critical thinking, understanding, and writing diminish when consistently offloaded to AI. As such, AI-independent literacy ensures individuals to retain the cognitive skills necessary to function when AI becomes suddenly unavailable. We note that AI-independent literacy is not about being efficient, but staying resilient under scenarios of AI Lock-In.

However, implementing such AI literacy education for individuals cannot be done without the support from organizations and nations. Recognizing that AI induced deskilling will collectively lead to systematic dependency (Section 4), organizations and nations must invest in AI literacy education through funding and public awareness campaigns. Through AI literacy, individuals can use AI with informed awareness, as well function without the use of AI, preserving their own cognitive autonomy in the age of AI Lock-In.

### 5.2. Bet-Hedging as a Long Term Thriving Strategy for Organizations

Organizations are embedding AI into their workflows and gradually replacing entry-level workers as a cost-cutting measure. However, this approach (i) increases the risk of operational failures and legal liabilities due to AI-induced errors and hallucinations, (ii) reduces opportunities to hire and train junior employees who could identify and address such issues, and (iii) ultimately disrupts the pipeline of human capital by producing fewer experienced professionals, locking the organization into structural AI dependency.

As such, rather than replacing entry-level workforces with AI, we urge organizations to consider *bet-hedging* on AI incorporated workflows: a strategy used in evolutionary biology where organisms intentionally sacrifice optimal efficiency to maintain diverse capabilities, ensuring long term survival and thriving (Seger & Brockman, 1987). Bet-hedging may initially seem costly and irrational as described in Section 2, but the koala's path toward extinction exempli-

fies what happens when bet-hedging is absent.

Specifically, bet-hedging on AI workflows for entry-level positions can be conceptualized based on three tiers of responsibility: (i) AI is mainly used on initial production tasks such as drafting, data collection, and simple routine tasks; (ii) junior employees focus on reviewing outputs from AI, checking factual errors, assessing alignment with organizational context; and (iii) senior professionals makes the final approval while providing juniors with feedback to help them develop judgment that AI cannot codify.

We acknowledge that this bet-hedging strategy reduces the short-term efficiency and incurs additional costs, which may be challenging for organizations facing budget limitations or pressure under competitive markets. However, it directly addresses the two pathways of organizational AI Lock-In identified in Section 3: juniors embedded in AI workflows can find errors and hallucinations before they lead to larger harm, while the continuous transfer of tacit knowledge from seniors to juniors sustains a pipeline of skilled professionals capable of making independent decisions when AI becomes unavailable. In essence, bet-hedging can serve as a safeguard against what Bainbridge (1983) called the *ironies of automation,* the paradox that as automated systems take over routine work, the humans expected to oversee them gradually lose the competence to do so. Most importantly, the bet-hedging approach is only possible while we remain a skilled society. A deskilled society will lack both the capable juniors needed to verify AI outputs and the experienced seniors needed to guide them, reinforcing why intervention must begin now.

### 5.3. Nation Should Prioritize AI Resilience

While valuable, the measures proposed for individuals and organizations remain voluntary and are thus limited in reach. Nations, however, have the regulatory authority to mandate compliance at scale, and with that authority comes a duty to

safeguard society against AI Lock-In.

To this end, we call for regulations that strengthen national **AI resilience** to nations actively incorporating AI into society, ensuring that society maintains its essential functions when AI becomes suddenly unavailable. Specifically, we ask: (i) mandatory audits over critical infrastructures such as power grids, healthcare systems, and government agencies, verifying their ability to maintain core functions during AI outages, (ii) incentivizing organizations that continue to hire and train junior employees through subsidies that offset the short-term costs of keeping human capital; and (iii) periodic AI-free drills, namely scheduled disconnections from AI services, to stress-test societal readiness and to provide awareness of such danger.

We recognize that such proposals to regulate AI may appear counterintuitive at a time when nations worldwide are racing to accelerate AI adoption (Maslej et al., 2025). Moreover, conducting resilience audits, providing organizational incentives, and executing AI-free drills at a national scale will involve huge economic and societal costs. However, we believe that these preemptive investments will be far smaller than the costs society will bear once AI Lock-In has fully taken hold. Earthquakes are infrequent, yet societies worldwide conduct regular drills precisely because when such a rare event occurs, deliberate practice helps us survive without panicking (Shaw et al., 2004). These measures serve the same function: they ensure that when AI disruption occurs through accidents, cyber attacks, or geopolitical coercion (Section 4), society is not caught helpless by an event it knew was possible but for which it never rehearsed.

## 6. Alternative Views

In this section, we present alternative perspectives that challenge the central arguments of our position.

**A1. Recurring Pattern of Technological Anxiety.** The topic of 'adverse effects of new, emerging technology on humanity' has been a recurring and debated subject throughout human history. From the Spinning Jenny to radio, GPS, and smartphones, each new technology has been accompanied by discourse claiming that it would make humans intellectually weaker or undermine their autonomy (Eisenstein, 1980; Postman, 2011; Carr, 2008). However, despite warnings, humanity has adapted well to these new technologies. This raises a question: Is the current claim on AI Lock-In just *old wine in new bottles?*

**R1.** We argue that the current AI Lock-In is not *old wine in new bottles*, which shows qualitatively distinct characteristics compared to past technological inventions.

First, at the individual level, current deskilling induced by AI is different from that of prior technologies in the extent to which it disrupts human autonomy in cognitive judgment (Kosmyna et al., 2025). Earlier tools, such as the calculator or GPS, replaced only a single function of our cognitive skills (*e.g.,* the calculator for arithmetic, the GPS for navigation) while leaving the higher-order judgment intact (*e.g.,* when and how to use them). For instance, when four diners want to split a $100 bill, individuals have to recognize that arithmetic is needed, formulate the calculation, and type it into the calculator (*i.e.,* $100 / 4). The tool executed the computation, but the framing and comprehension of the situation remained our task. However, AI is troubling as it simply abstracts the entire process into a single line of prompt, *four of us had dinner, and the bill is $100. How much do I owe?* In the past, the judgment of how to understand, interpret, and determine what needs to be done to resolve a situation remained within the human domain. Now, AI is replacing not merely a single function of our cognitive ability, but our autonomous cognitive agency itself.

Second, at the societal level, the speed at which AI is penetrating and disrupting existing occupations is unprecedented in both its pace and its breadth across sectors, leaving humanity with drastically less time to adapt than past automation allowed. In the past, automation and the resulting displacement of jobs happened sequentially, allowing time for adjustment: as automobile assembly lines became mechanized, new roles emerged for designing and maintaining the machinery itself, justifying the need for entry-level hiring. However, the current job displacement with AI is occurring indiscriminately and across roles that were once considered the exclusive province of human intelligence, including creative, strategic, and analytical work. Unlike past technologies that displaced workers in one domain while opening opportunities in another, replacement with AI is providing a short timeline for adjustment and the alternatives needed for displaced workers.

**A2. Development of Technology.** One of the concerns of AI Lock-In lies in the possibility of sudden AI disconnection. While the scenarios illustrated in this paper are plausible, and similar events have occurred in the past, what if technological advances enable reliable AI access anywhere and at any time? For instance, on-device AI could offer a promising solution (Wang et al., 2025). On-device AI runs models directly on local hardware without requiring an internet connection, making such systems inherently robust against cyber attacks, API service disruptions, and geopolitical restrictions on cloud-based services. If on-device AI becomes widely available, sudden AI disruption may pose far less of a systemic threat, as users and organizations could simply switch to locally hosted alternatives. Indeed, on-device AI currently faces limitations: lower performance compared to server-based models, and strict hardware and memory constraints that restrict deployable models. However, given the current pace of AI and technology development, these

problems are likely to be solved in the near future.

**R2.** We acknowledge the value of these technical directions and view them as complementary to our position. However, even under the idealized scenario where AI can be constantly accessed without any disruption, the central concern of our position is not addressed: cognitive offloading is leading to human deskilling. Doctors will lose the ability to diagnose on their own, and students will struggle to solve problems without assistance, not because AI has failed them, but because it never did. Some may question: Does it matter? Our position is that it does, because it undermines the human autonomy to think, plan, and make decisions. Just as the invention of the automobile did not stop us from walking, we must not let AI erode our capacity to think for ourselves.

## 7. Discussion and Limitation

**User Agency Matters.** As the philosopher Karl Jaspers argued, the magnitude of dependence on technologies varies across individuals, and its impact differs depending on how they are used. Critical users maximize the positive aspect of new technologies, while passive users are exposed to more negative effects. In other words, the impact of technology cannot be viewed in a linear or deterministic manner.

**Need for Adult Re-education.** We argue that to address individual-level AI Lock-In, AI literacy education should be expanded beyond the K-12 curricula to adult re-education. This extension is urgently needed for adults who have never received such training in their formal schooling, yet are already frequently using AI. Unlike younger generations who may have AI literacy as part of their education (OECD, 2025), most working adults are using AI tools without safeguards, making them particularly vulnerable to cognitive offloading and deskilling. Providing AI literacy education to adults requires the support of organizations and nations.

**Investment on Sovereign AI.** While Sovereign AI is not the solution for AI Lock-In, we believe that such an initiative could be part of the bet-hedging strategy that nations can implement. As we have noted earlier that a nation that has achieved Sovereign AI is still vulnerable to AI Lock-In (Section 4), such an initiative still alleviates problems arising from vendor Lock-In risk by diversifying the sources of AI infrastructure and services that can be leveraged in geopolitical coercion. In summary, our position is that Sovereign AI does not necessarily solve AI Lock-In risk, but it mitigates the geopolitical vulnerabilities associated with foreign AI dependency and cloud platforms.

**Distinguishing AI Lock-In from Vendor Lock-In.** We strongly believe that vendor Lock-In is an equally important problem that requires attention from all stakeholders in society. However, while vendor Lock-In is about being locked to a specific AI provider, AI Lock-In is about being overly dependent on AI itself. Briefly, solving vendor Lock-In does not solve AI Lock-In, but solving AI Lock-In can solve vendor Lock-In on AI services. As such, AI Lock-In is the more fundamental problem, while Vendor Lock-In is a separate risk that, when it overlaps with AI Lock-In, it can have a compounding effect. We believe that both are important problems, but we intentionally focused on the root challenge to maintain the clarity of our position.

**Limitation.** Unfortunately, the scope of our position is limited by the uneven global distribution of AI development (Maslej et al., 2025; Appel et al., 2025). The use of AI has been concentrated on a handful of nations with strong digital infrastructure and capital, similar to historical patterns of technology distribution. For developing countries without much AI infrastructure and access, the discussion of AI Lock-In and AI literacy may seem like privileged concerns with little immediate relevance.

Moreover, the correlational nature of the provided evidence is unavoidable given the phenomenon we are describing. As we have stated in our title, *AI Lock-In is in progress*, AI Lock-In is an emerging and ongoing process. As such, the full trajectory of AI Lock-In does not yet exist. This is similar to how the early literature (Hansen et al., 1981) on climate change has approached, where initial warnings necessarily relied on correlational and observational studies. We believe that waiting for such evidence to accumulate would itself deepen the risk we are warning against.

## 8. Conclusion

Technology has made our lives more convenient, but this convenience may come at the cost of individuals' and societies' autonomy. In this position paper, we argued that society now stands at a critical juncture: the rapid and somewhat indiscriminate adoption of AI into our everyday lives is pushing the current skilled society to transition into a deskilled one. Once this threshold is crossed, we warn that individuals, organizations, and nations will be AI Locked-In: a state at which our reliance on AI becomes so deeply embedded, eroding human capability for autonomous thought and skilled action. Yet, we argue that humanity has the foresight to recognize this trajectory we are on and the agency to alter it. The path we advocate is not to reject AI, but to ensure that AI adoption proceeds alongside the growth of human resilience against AI dependency. Collective effort from all stakeholders: individual, organization, and nation is necessary. We hope that our work can bring AI Lock-In into more broader public awareness, and inspire researchers and policy makers to work on these challenges together.

## Acknowledgements

We thank the ICML reviewers for their comments and suggestions. This work is supported by the National Research Foundation of Korea (NRF) grant funded by the Korea government (MSIT) (No. RS-2024-00358602) and the Institute of Information & Communications Technology Planning & Evaluation (IITP) grant funded by the Korea government (MSIT), Artificial Intelligence Graduate School Program (No. RS-2019-II190079, Korea University), the Artificial Intelligence Star Fellowship Support Program to nurture the best talents (No. RS-2025-02304828), and the AI Research Hub Project (No. RS-2024-00457882), and Development of Unified Reasoning Technology Mimicking Human Cognition for Hierarchical Understanding and Unbounded Problem Solving (No. RS-2026-25522672).

The authors would like to acknowledge Ethan Cho, whose earlier reflections shared on LinkedIn inspired part of the motivation and early thinking behind this position paper. We also thank Guk Jo, Hyunjong Yu, and Keonwoo Kim for reviewing our manuscript and providing valuable feedback.

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

# A. Lock-In Case Studies

In this section, we present two case studies that illustrate how technology-induced lock-in in the past led to measurable harm, and how institutions attempted to address the resulting consequences. These case studies are intended to demonstrate that the mechanisms underlying AI Lock-In, as discussed in this paper, are not speculative. Rather, they have well-documented historical precedents that can both motivate and inform how we address AI Lock-In going forward.

**Case 1. Cockpit Automation and Pilot Deskilling.**

The automation of commercial aviation cockpits has provided substantial safety and efficiency benefits over decades. However, this progressive automation has also been accompanied by a gradual erosion of pilots' manual flying skills. In a domain as safety-critical as aviation, where advanced technology directly safeguards human lives, this erosion of human capability has led to catastrophic consequences. The crash of Air France Flight 447 on June 1, 2009, is a case in point.

The crash killed all 228 people on board. When ice crystals blocked the aircraft's pitot tubes at cruising altitude, the speed indications became unreliable and the autopilot disconnected. The Bureau d'Enquêtes et d'Analyses (BEA) investigation concluded that none of the three crew members was able to recover the aircraft, citing *the crew's failure to diagnose the stall situation and, consequently, the lack of any actions that would have made recovery possible* (Bureau d'Enquetes et d'Analyses pour la securite de l'aviation civile, 2012). Notably, this was not an isolated crew failure: the Federal Aviation Administration (FAA) subsequently recognized the systemic nature of this risk, stating that *continuous use of autoflight systems could lead to degradation of the pilot's ability to quickly recover the aircraft from an undesired state* (Federal Aviation Administration, 2013). The accident illustrates the deskilling mechanism described in Section 3: prolonged cognitive offloading to automated systems eroded the crew's capacity to perform independently when the system became unavailable.

The aviation community responded with a multi-layered institutional framework explicitly designed to counteract automation-induced deskilling:

- The BEA investigation recommended increased manual flying in pilot training, improved basic airmanship instruction, and simulator scenarios exposing pilots to high-stress situations with sudden automation disengagement (Bureau d'Enquetes et d'Analyses pour la securite de l'aviation civile, 2012).

- The European Aviation Safety Agency (EASA) issued Safety Information Bulletin 2013-05, noting that "continuous use of auto-flight systems could lead to potential degradation of the pilot's ability to cope with the manual handling of the aeroplane," and recommending that operators incorporate manual flight practice into both initial and recurrent training as well as, when feasible, line operations (European Union Aviation Safety Agency, 2025).

- FAA Safety Alert for Operators (SAFO) 17007 strengthened these recommendations, requiring training programs to incorporate manual flight proficiency exercises including upset recovery, stall prevention, go-arounds, and visual approaches without full automation (Federal Aviation Administration, 2017).

**Case 2. Platform Monopoly Lock-In: E-Commerce and Consumer Dependency.** While the aviation case illustrates skill-level lock-in among specialized professionals, platform monopoly lock-in demonstrates how dependency can emerge at a population scale through the absence of viable alternatives. Modern e-commerce ecosystems exemplify this dynamic. Amazon has built an integrated ecosystem, encompassing logistics infrastructure, streaming services, smart home devices, and cloud computing, which creates multi-layered switching costs for consumers.

Amazon's dominant market position has not insulated its users from harm. If anything, the absence of viable alternatives has enabled the platform to absorb repeated, large-scale privacy violations with minimal consequences to its user base. In 2021, the Luxembourg National Commission for Data Protection (CNPD) imposed a €746 million fine for processing users' personal data for targeted advertising without valid consent (Leggett, 2021). In 2023, the Federal Trade Commission (FTC) and Department of Justice charged Amazon with violating the Children's Online Privacy Protection Act, finding that the company retained children's voice recordings collected through its Alexa voice assistant indefinitely and used the data to train its algorithms, while failing to honor parents' deletion requests (Federal Trade Commission, 2023). Despite such substantial violations, Amazon's Prime membership has continued to grow steadily, surpassing 200 million members in the United States alone (Consumer Intelligence Research Partners, 2025). This persistence of dependency despite documented harm illustrates a key mechanism of lock-in: when a platform has achieved sufficient ecosystem integration and the switching costs exceed the perceived risk, rational individual action cannot resolve the collective vulnerability.

These two case studies together demonstrate that the mechanisms of AI Lock-In we identify are neither novel nor speculative. What is novel is the convergence of both dynamics in AI systems: generative AI simultaneously induces cognitive deskilling (as in aviation) and creates ecosystem-level structural dependency  as in e-commerce), operating across virtually all professional and personal domains at once.

## B. Operationalizing AI Literacy in Scale based on Media Literacy

We provide a detailed operationalization of our proposal regarding the AI literacy education. We propose the following: (i) designing a structured AI literacy workshop based on the five core concepts in Table 2, and (ii) proposing a multi-level AI Lock-In Index for systematically monitoring AI dependency at micro and macro levels.

### B.1. Structured AI Literacy Workshop

In Section 5.1, we adapted the Center for Media Literacy's Five Core Concepts to AI literacy education (Table 2). Here, we describe how each concept can be operationalized as a concrete workshop exercise, forming a unified AI literacy program deployable across schools, organizations, and national education initiatives.

**Exercise 1: Source Verification Habits (Authorship).** Participants are trained to systematically request and verify original sources for AI-generated outputs. When querying AI for factual information, participants practice asking for source references and cross-checking them against primary materials. This exercise draws on a historical parallel: the transition from print to the internet introduced new risks of encountering unverified information, which were partly addressed by institutionalizing source verification, as exemplified by Wikipedia's policy requiring all claims to be supported by reliable sources. Similarly, cultivating the habit of source verification for AI outputs trains users to treat AI responses as hypotheses to be checked rather than authoritative answers.

**Exercise 2: Confidence Calibration (Format).** Participants are presented with AI responses that are confidently worded but factually incorrect. They are asked to evaluate the accuracy of each response before the correct answer is revealed. This exercise exposes the *illusion of competence*, the tendency for AI's fluent and authoritative tone to instill unwarranted trust, and trains participants to decouple linguistic fluency from factual correctness. Such calibration exercises can be designed with varying difficulty across domains (*e.g.,* general knowledge, science, current events) to suit different audiences.

**Exercise 3: Filter Bubble Comparison (Audience).** In this group exercise, multiple participants pose the same question to the AI platform and then compare their respective outputs. Because each participant's individual prompting habits, phrasing, and conversational history can shape the AI's response, this side-by-side comparison makes visible the filter bubbles that AI systems can create around individual users. Participants discuss how differences in input framing led to divergent outputs, developing awareness that AI responses are not universal but are shaped by user-side factors.

**Exercise 4: Perspective Gap Analysis (Content).** Participants query AI on culturally sensitive or globally relevant topics and examine which perspectives are included or omitted in the responses. For example, asking an AI system to "describe a typical breakfast" tends to yield descriptions of Western dishes such as pancakes and bacon, while omitting equally common breakfast traditions from other cultures. By systematically identifying such gaps, participants develop the ability to critically evaluate cultural biases inherent in AI training data and recognize that AI outputs reflect the distribution of its training corpus rather than an objective representation of the world.

**Exercise 5: Engagement Design Awareness (Purpose).** Participants examine how AI services are designed to sustain user engagement; for instance, ending every response with a follow-up question to encourage continued interaction. By identifying such patterns, participants develop awareness that their usage behavior can be shaped by commercial design choices rather than their own needs.

**Workshop Integration.** These five exercises are designed to be delivered as a single, structured workshop program rather than standalone activities. The progression moves from verifying individual AI outputs (Exercises 1-2) to understanding how AI responses vary across users and cultures (Exercises 3-4), and finally to examining the systemic design choices behind AI services (Exercise 5). This structure enables participants to build AI-engaged literacy (*i.e.,* the ability to critically interact with AI), which in turn strengthens their awareness of the risks described in our AI Lock-In framework.

# C. Role of AI Researchers

As AI researchers, we wrote this position paper with a responsibility that we are among the first to witness AI Lock-In firsthand, and thus among the first obligated to sound the alarm, before intervention becomes too late. To prevent AI Lock-In, we outline what the AI research community can contribute and should further develop. We first provide a high-level overview of what needs to be done.

**1. Measurement.** We need to understand the current state of AI Lock-In. Such an objective can be achieved by developing an AI Lock-In Index for international and national organizations to periodically monitor AI dependency. Development of benchmarks that can assess deskilling is also recommended.

**2. Designing.** We need to consider AI Lock-In as an important alignment factor that needs to be considered in model designing. Many alignment works focus on user retention and safety, but often dismiss *how much we are preserving human agency*. Such technical research directions need to be supported by organizations and nations.

**3. Testing.** We need testing frameworks that can assess how much individuals and organizations are dependent on AI. Similar to red teaming research in adversarial robustness, a similar approach can be considered.

We provide further details on how we can further develop AI Lock-In for measurement in the following section.

## C.1. Development of Multi-Level AI Lock-In Index as a Measurement Tool

We propose the development of the **AI Lock-In Index**: a standardized measurement framework designed to enable international and national organizations to periodically monitor AI dependency progression. The index captures dependency at two complementary scales: a micro-level individual dependency and a macro-level structural dependency, recognizing that AI Lock-In operates simultaneously across both dimensions as described in Section 3.

### C.1.1. MICRO-LEVEL: INDIVIDUAL DEPENDENCY MEASUREMENT

At the individual level, the AI Lock-In Index measures the degree to which individuals have become cognitively dependent on AI systems. This layer combines two complementary instruments:

**Self-Report Survey.** A standardized questionnaire measuring key dimensions of individual AI dependency, including:

  (i) **AI reliance frequency**: how often individuals defer to AI for tasks they could perform independently;

 (ii) **AI-independent confidence**: self-assessed ability to complete tasks without AI assistance;

(iii) **Preemptive AI dependency**: the tendency to consult AI before attempting independent thought or problem-solving.

Items are measured on 7-point Likert scales and designed for longitudinal tracking of dependency changes over time.

**Metacognitive Calibration Test.** A behavioral assessment in which participants evaluate the accuracy of AI-generated responses across a set of domain-relevant questions. The gap between participants' trust in AI accuracy and the actual accuracy of the AI's responses serves as a measure of metacognitive calibration, or miscalibration, indicating overreliance. This test complements self-report data by capturing dependency patterns that individuals may not consciously recognize.

### C.1.2. MACRO-LEVEL: STRUCTURAL DEPENDENCY MEASUREMENT

At the organizational and national level, the AI Lock-In Index assesses how deeply AI has become embedded into institutional workflows and critical infrastructure, capturing the systemic dimension of AI Lock-In that compounds individual-level dependency.

**Organizational Indicators.** Key metrics include:

  (i) **AI-embedded workflow proportion**: the share of core business processes that rely on AI systems for routine operation;

 (ii) **Workforce substitution rate**: the extent to which roles previously filled by human workers have been replaced or augmented by AI;

(iii) **AI outage impact**: measured operational downtime or productivity loss during AI service disruptions;

(iv) **Human capital pipeline health**: entry-level hiring rates and junior-to-senior progression ratios, as indicators of whether tacit knowledge transfer (discussed in Section 3.2) is being maintained.

**National Indicators.** Key metrics include:

(i) **Critical infrastructure AI reliance**: the proportion of essential services (*e.g.,* power grids, healthcare systems, government administration, transportation) that depend on AI for core functions;

(ii) **AI service provider concentration**: the degree to which a nation's AI usage is concentrated among a small number of providers (*e.g.,* a single API provider accounts for a dominant share of national AI usage);

(iii) **AI-free fallback availability**: whether critical systems maintain operational capability without AI, as recommended in Section 5.3.

### C.1.3. FROM INDEX TO ACTION

The AI Lock-In Index is designed not merely as a diagnostic tool but as a basis for policy intervention. By tracking micro-level and macro-level indicators over time, analogously to how GDP tracks economic output or the Human Development Index tracks societal well-being, the index enables governments and organizations to identify when AI dependency is approaching critical thresholds and to implement the resilience measures proposed in Section 5, including AI literacy programs Section 5.1, organizational bet-hedging strategies Section 5.2, and national AI resilience regulations Section 5.3.

