# OpenReview forum: "Position: AI Lock-In Is in Progress, and We Must Be Prepared"
_ICML.cc/2026/Position_Paper_Track — ICML 2026 Position Paper Track spotlight_

### Official Review · Reviewer_uiGd · 2026-02-21

**Significance:** 3
**Argument Clarity:** 4
**Rating:** 5
**Confidence:** 4

**Questions:**

Personally speaking, this position is very intriguing and practical in my review batch. However, the current supporting evidence and concrete descriptions need augmentation. **I initially rate this position paper as Borderline Reject, and I look forward to updating the rating based on the authors' response and discussion.**

**Alternative Views Section:**

Yes

**Compliance With Llm Reviewing Policy A Conservative:**

Affirmed.

**Discussion Potential:**

3

**Final Justification:**

I have read the authors' response. It is convincing and comprehensive. I believe the topic of AI Lock-in needs community attention, and this position paper is timely. My final rating is raised to Accept.

**Paper Summary:**

The paper proposes the position that AI safety must address AI Lock-In (aka, societal dependency that deskills humans) and that we should invest in AI resilience via dual-track literacy and national AI-free drills. The evidence is from productivity claims with selected examples (e.g., StackOverflow decline), and the authors call for early intervention.

**Position:**

Yes

**Position In Title:**

Yes

**Related Work:**

2

**Strengths And Weaknesses:**

**Strength**
1. The paper is well-formatted, well-written, and the recommended actions align with the position.
2. The research topic of dependency risk and resilience is timely and important.
3. The technical terms are clearly explained and defined in the paper. The provided evidence in Figure 1 is relevant.

**Weakness**
1. The proposed position is mostly the narration of “automation -> deskilling” concerns, which have been long discussed ever since the invention of the Spinning Jenny in the 1760s. What would be the specific new insight of deskilling regarding the invention of generative foundation models? That is, how are the specific patterns of deskilling different between the 1760s and 2020s? Please note that this also naturally connects to the societal impacts caused by AI alignment [1,2,3].

2. The used evidence (though clear) is not sufficient: many findings from the evidence are correlational without specific quantitative metrics.

3. While the current title is very concise and clear, please revise the header from Page 2 to Page 11. It seems to be the old version of the title, which is indeed not as good as the current version.

[1] Ji et al., AI Alignment: A Comprehensive Survey. https://arxiv.org/abs/2310.19852

[2] Yang et al., Reliable and Responsible Foundation Models. https://openreview.net/forum?id=nLJZh4M6S5

[3] Huang et al., On the Trustworthiness of Generative Foundation Models: Guideline, Assessment, and Perspective. https://arxiv.org/abs/2502.14296

**Support:**

2

---

> ### Author Rebuttal · Authors · 2026-03-31
>
> Thank you for the positive remarks on our work and the willingness to engage in constructive discussion. We have prepared detailed responses to each of the reviewer's concerns below.
>
> **W1. How AI-Deskilling Differs from Past Events**
>
> >**[A1]** First of all, we would like to thank the reviewer (alongside all other reviewers) for this question, where we have used this to enforce the **Alternatives** section. We now provide counter arguments describing why the current AI-Lock In is simply not *old wine in new bottle*. We explain the two main distinctions below.
>
> > **First, the deskilling has shifted from physical execution to cognitive tasks, making AI-induced deskilling more consequential.** Historical automation, such as the Spinning Jenny, replaced domain-specific physical labor. When these machines broke down, the malfunction was immediately visible, and its consequences were bound. A broken machine might disrupt textile production, but little else. However, AI is replacing cognitive tasks [1] in high-stakes domains such as medical diagnosis, where errors are neither easily detectable nor confined in their impact. For instance, patients may require timely and accurate clinical judgment, but with AI-induced deskilling, physicians may no longer retain the full capacity to diagnose independently. Unlike machine failures, such cognitive atrophy remains invisible until the moment it matters most.
>
> > **Second, AI-induced deskilling is self-undermining in a way that prior automation was not.** The alignment literature cited by the reviewer is premised on the assumption that humans retain sufficient cognitive capacity to supervise AI systems. For instance, Ji et al. (2023) explicitly identify *controllability* as a core alignment objective, defined as ensuring that AI systems "remain subject to human oversight and intervention." However, as AI Lock-In deepens and deskilling progresses, we argue that this assumption may not hold. The progressive offloading of cognitive tasks to AI leads to reduced capability of human oversight [1]. As such, the more society relies on AI, the less likely it is to oversee it. This is not what we have observed in past historical automation, where human cognitive skills remained even as physical labor was displaced. Based on this, we have added the following counterargument to the Alternatives section:
>
> > *"... alignment frameworks (Ji et al., 2023; Yang et al., 2025; Huang et al., 2025) and regulations such as the EU AI Act's human oversight provisions (Article 14; European Parliament, 2024) assume that we will have (and maintain) the capacity to oversee and supervise AI operations. However, we argue that the current offloading of critical reasoning to AI is undermining ...*
>
> **W2. Quantitative Metrics**
>
> >**[A2]** We first acknowledge that some of the evidence we present is correlational in nature. We believe this is somewhat inevitable as we are studying deskilling in human subjects, where designing fully controlled causal experiments on cognitive atrophy is ethically and practically difficult. However, we have made our best efforts to include the strongest available causal and quantitative evidence at the time of writing. Specifically, the Budzyn et al. (2025) study on adenoma detection rates and the Bastani et al. (2025) study on mathematics performance both employ before-and-after designs that move meaningfully beyond simple correlation. In response to the reviewer's concern, we have additionally incorporated findings from a recent study on how AI impacts skill formation in coding [2], which provides further numerical grounding.
>
> > Moreover, the correlational nature of some evidence is unavoidable given the phenomenon we are describing. As we have stated in our title, "AI Lock-In is in progress", this is an emerging and ongoing process. As such, the full trajectory of AI Lock-In does not yet exist. This is similar to how the early literatures [3] on climate change have approached, where initial warnings necessarily relied on correlational and observational studies. We believe that waiting for such evidence to accumulate would itself deepen the risk we are warning against.
>
> >As AI researchers **we wrote this position paper with a responsibility that we are among the first to witness these dynamics firsthand, and thus among the first obligated to sound the alarm**, before intervention becomes too late.
>
> **W3.Running Title**
>
> >**[A3]** Thank you for catching this inconsistency. We have updated accordingly.
>
> [1] Your brain on ChatGPT: Accumulation of cognitive debt when using an AI assistant for essay writing task (ArXiv, 2025)
>
> [2] How AI Impacts Skill Formation (Anthropic, 2026)
>
> [3] Climate Impact of Increasing Atmospheric Carbon Dioxide (Science, 1981)
>
> We hope that our reply has answered the questions raised by the reviewer. We are happy to engage in further discussions if additional concerns remain. Thank you once again for your time and dedication.

---

> > ### Author Rebuttal · Reviewer_uiGd · 2026-04-03
> >
> > I appreciate the authors' responses, which are very detailed and convincing. I especially enjoy the discussion on cognitive tasks and self-undermining.

---

### Official Review · Reviewer_fGWy · 2026-03-10

**Significance:** 3
**Argument Clarity:** 3
**Rating:** 5
**Confidence:** 3

**Questions:**

1. Under what conditions does individual-level cognitive offloading translate into systemic vulnerability rather than remaining a personal productivity trade-off? Can you provide a more rigorous account of the aggregation mechanism from individual to national Lock-In?

2. The threat scenarios assume correlated, simultaneous AI failure across diverse systems. How do you assess the probability of such failure given heterogeneous providers and architectures? Does this heterogeneity provide natural resilience the paper underweights?

3. Bet-hedging and AI-free drills face collective action problems: early adopters bear costs while competitors do not. What incentive structures would make these proposals sustainable in competitive markets?

4. What is the principled argument for why AI dependency is qualitatively different from historical dependencies on electricity or the internet, which triggered analogous deskilling fears?

5. Could decentralized AI infrastructure (federated models, open weights, local deployments) substantially mitigate the systemic risks you identify, weakening the need for the proposed policy interventions?

**Alternative Views Section:**

Yes

**Compliance With Llm Reviewing Policy A Conservative:**

Affirmed.

**Discussion Potential:**

3

**Final Justification:**

The authors have provided thorough and well-structured responses that adequately address my concerns.

**Paper Summary:**

This paper introduces the notion of *AI Lock-In*, adapting the economic concept of technological lock-in to argue that society is progressively becoming so dependent on AI that reverting to human-centered alternatives is growing prohibitively costly. The authors draw a parallel with the koala's evolutionary specialization on eucalyptus, which initially boosted efficiency but ultimately left the species unable to adapt when conditions changed.

The paper traces how AI Lock-In manifests at three levels: (1) *individual* — cognitive offloading leads to skill atrophy, evidenced by declining StackOverflow usage, reduced colonoscopy detection rates after AI-assisted training, and degraded student math performance upon AI removal; (2) *organizational* — firms replace entry-level workers with AI, eroding tacit knowledge pipelines and creating operational fragility; and (3) *national* — governments accelerate AI adoption for competitive advantage, creating strategic dependencies vulnerable to accidental failures, cyber attacks, and geopolitical coercion.

To counter these risks, the authors propose three interventions: AI literacy education that cultivates both AI-engaged and AI-independent competence for individuals; a "bet-hedging" strategy for organizations that intentionally preserves human roles AI could fill; and national "AI resilience" regulation including mandatory infrastructure audits and periodic "AI-free drills."

**Position:**

Yes

**Position In Title:**

Yes

**Related Work:**

3

**Strengths And Weaknesses:**

**Strengths:**

- Grounding the argument in economic lock-in theory (Arthur, 1989) and the koala specialization analogy (Table 1) gives the paper a coherent, memorable analytical framework accessible to a broad audience.

- The multi-level analysis is systematic, and the individual-level evidence is particularly strong: StackOverflow traffic decline (Figure 1), colonoscopy deskilling (Budzyń et al., 2025), and math tutoring rebound (Bastani et al., 2025) are well-chosen examples.

- Section 5 proposes specific, differentiated interventions at each level. The AI-engaged vs. AI-independent literacy distinction and the "AI-free drills" proposal are genuinely useful conceptual contributions.

- The paper identifies a real gap in AI safety discourse: the field focuses on alignment and harm prevention while largely overlooking dependency risk. This reframing is valuable and timely.

**Weaknesses:**

- The causal chain from individual deskilling to national-level Lock-In involves under-argued leaps. The aggregation mechanism is asserted rather than modeled, and the organizational/national levels rely more on speculation than on the empirical grounding that strengthens the individual-level analysis.

- The disruption scenario on page 5 (traffic lights, logistics collapse, government shutdown) assumes simultaneous correlated failure across diverse AI systems without probability assessment, reading as speculative fiction rather than risk analysis.

- The "old wine in new bottles" counter-argument (Section 6) is acknowledged but hastily dismissed. The paper does not rigorously distinguish AI dependency from historical dependencies on electricity or the internet, and the extended cognition perspective (Clark & Chalmers, 1998) is mentioned but not engaged with.

- Notable omissions in related work: the human factors literature on automation complacency (Bainbridge's "ironies of automation"), Carr's "The Glass Cage," and the EU AI Act's human oversight provisions.

- The bet-hedging and AI-free drills proposals face collective action problems that the paper acknowledges but does not adequately address, yet these incentive barriers may be the binding constraint on any proposed intervention.

**Support:**

3

---

> ### Author Rebuttal · Authors · 2026-03-31
>
> We thank the reviewer for evaluating our work as *identifying a real gap in AI safety discourse*, and commenting that the paper is *valuable and timely*. Below, we have prepared detailed responses.
>
> **[W1, Q1.] Individual to Systemic Risk**
>
> >[A1] We argue that the transition from individual to collective risk emerges when organizations or nations enforce, or institutionalize AI usage as a performance target, thereby removing the individual's option on whether to use AI.
>
> > One recent example is Amazon deploying an internal monitoring system to check employees' AI usage, with the results being used for performance review [1]. In such a case where AI usage becomes a goal, users will no longer have the choice to offload tasks but will be forced to. A similar logic will work on the national level, where governments are incentivizing AI deployment (Section 3.2; National level). As such, an individual's cognitive offloading becomes systemic risk when institutions enforce or regulate the use of AI. Based on this, we have added the following in L173:
>
> > "... individual to systemic dependency is accelerated when organizations and nations actively incentivize or mandate AI usage as a performance target..."
>
> **[W2, Q2] Heterogeneity in Service as a Resilience, Probability Assessment**
>
> >[A2] We agree with the reviewer's view that heterogeneity across AI services will offer partial resilience under accidental events (e.g., earthquake). However, such resilience might not work under geopolitical coercion scenarios, where restrictions or attacks are made at an institutional or national level. As we argued in Sec 4.3, weaponizing AI Lock-In can become a highly probable scenario in the future.
>
> > Regarding the probability assessment of risks, we acknowledge that such quantitative risk analysis would be beneficial. However, much of the relevant information (e.g., dependency between services, accident rates) remains inaccessible. We believe that such an assessment would be necessary and should be developed. Here we leave it as an important future research direction.
>
> **[W3, Q4] Old Wine in New Bottle**
>
> >[A3] Due to limited word counts, we cordially ask the reviewer to refer to our response to Reviewer4 (uiGd)-A1. In short, we have included an in-depth discussion in our alternative sections to include counterarguments on why the current AI Lock-In is not  *old wine in a new bottle.*
>
> **[W4] Related Works**
>
> >[A4] Thank you for highlighting these references. We have incorporated: (i) Bainbridge's (1983) "ironies of automation" in L343 to justify bet-hedging as a safeguard against automation-induced skill degradation, (ii) Carr's work in L152 to support the claim that cognitive skills atrophy when offloaded to technology, and (iii) the EU AI Act's provisions (Article 14) in our counterargument to "Old Wine in New Bottles" (please kindly refer to our response to Reviewer4 uiGd-W1 for details).
>
> **[W5, Q3] How to Incentivize Bet-Hedging**
>
> >[A5] This is an important point, where we believe that the Bet-hedging strategy should be supported and incentivized with national-level regulations (Sec 5.3). This is similar to how workplace safety regulations and parental leave mandates incur short-term costs on organizations, but are sustained through financial incentives and tax subsidies that offset these costs in recognition of their long-term societal benefits. As explained in our work, nations should have the motivation to enforce this in scale to prevent AI Lock-In. Based on this, we have revised **Section 5.3** to explicitly include organizational incentives as a policy proposal alongside resilience audits and AI-free drills.
>
> **[Q5] Decentralized AI Infrastructure**
>
> >[A6] We acknowledge that decentralized AI infrastructures will be an important technical research direction that can reduce systemic risks. We view this as complementary to our position. However, even under the idealized scenario where AI can be constantly accessed without any disruption, the central concern of our position is not addressed: cognitive offloading is leading to human deskilling. Doctors will lose the ability to diagnose on their own, not because AI has failed them, but because it never did. Some may question: Does it matter? Our position is that it does, because it undermines human autonomy to think, plan, and make decisions [2]. Just as the invention of the automobile did not stop us from walking, we must not let AI remove our capacity to think for ourselves. AI Lock-In Risk, therefore, is not only about what happens when AI stops working, but also about what is lost when it works too well.
>
> [1] Amazon is determined to use AI for everything (Guardian, 26)
>
> [2] Your brain on ChatGPT: Accumulation of cognitive debt when using an AI assistant for essay writing task (ArXiv, 2025)
>
> We are very happy to receive these valuable comments, and would like to appreciate the reviewer. If any other concerns remain, please kindly let us know.

---

> > ### Author Rebuttal · Reviewer_fGWy · 2026-04-03
> >
> > The authors have provided thorough and well-structured responses that adequately address my concerns. 👍

---

### Official Review · Reviewer_gPWu · 2026-03-10

**Significance:** 3
**Argument Clarity:** 4
**Rating:** 5
**Confidence:** 4

**Questions:**

In the introduction, the stated position is "this position paper argues that AI safety research and regulation must address AI Lock-In," but the adaptations proposed to address AI lock-in seem to be entirely regulatory/policy based. What can AI safety researchers (or better yet, AI researchers in general) do to help address these issues?

In 4.1, you assert that wide adoption of open source software increases the risks of AI lock-in. Isn't this similar to other existing critical software infrastructure, e.g. the Linux kernel? How does the risk described in section 4.1 compare to the risks posed by critical infrastructure failure in other areas of technology or society at large?

In 4.2, are there examples of cyberattacks causing widespread issues due to lock-in that could be compared to? A generic example might be something like a hospital suffering a ransomware cyberattack and being unable to access patient health data due to a lack of offline or paper backups of that data, but a specific case study would be helpful here.

In 4.3, is is reasonable to analogize this to things like supply chain disruption due to offshoring of manufacturing or other key industries? If so, such disruptions would be strong motivation to address geopolitical/vendor AI lock-in risks as well.

In table 2, how can some of these questions be answered given that major commercial LLMs seeing high adoption rates are notoriously closed about the nature of the model and the data it is trained on? For example, it's hard to interrogate cultural bias and ethical assumptions embedded in the model if the training data and finetuning process is completely obscured from the public, including AI safety researchers.

Regarding the alternative views, what about the alternative that the downsides of AI lock-in can be avoided with sufficiently robust and reliable infrastructure, such that downtimes are minimal? This seems to be the attitude taken for a lot of key server/cloud infrastructure, and while it has natural downsides, can we dismiss the idea of it working acceptably in practice for AI as well?

**Alternative Views Section:**

Yes

**Compliance With Llm Reviewing Policy A Conservative:**

Affirmed.

**Discussion Potential:**

3

**Final Justification:**

The rebuttal addressed some of my concerns and added some clarification/argumentation to the paper to resolve the other ones substantively.

**Paper Summary:**

This paper argues that AI lock-in, the phenomenon of individuals/organizations/nations becoming dependent on AI for tasks that would previously be performed by humans, is on the rise, and that steps must be taken to avoid AI lock-in. The paper discusses how lock-in develops at different scales (individual/organizational/national) and the risks that each pose, and then presents a series of policy recommendations to mitigate these risks.

**Position:**

Yes

**Position In Title:**

Yes

**Related Work:**

3

**Strengths And Weaknesses:**

This paper concerns a relatively under-discussed AI risk, termed AI lock-in, which seems both relevant to the field and worthy of discussion. The paper is well written, with clear argumentation backing up the stated position.

In terms of weaknesses, the main issues I have are these three:

1. There's limited discussion of how lock-in has developed and caused harm in the past (which would help to motivate and inform how we address AI lock-in). Something like a case study would be useful here if one exists.
2. There's no counterargument or rebuttal to the alternate positions mentioned. In the words of the paper, IS this just old wine in new bottles? I'm not convinced that is the case, but there isn't a clear argument against that alternative, whether in the alternatives section or elsewhere.
3. While the stated position argues that AI lock-in is a concern for both AI safety research and AI regulation, the paper doesn't discuss ways that AI safety research can act to avoid AI lock-in or mitigate its harms.

That said, the paper as a whole is well written and provides evidence for its arguments, plus the topic is likely to spur productive debate in the field, so I am inclined to recommend acceptance. I do think that addressing the above weaknesses (especially #2) would greatly strengthen the paper, however, and I would raise my score if these three issues can be addressed.

**Support:**

3

---

> ### Author Rebuttal · Authors · 2026-03-31
>
> We are both happy and encouraged to see that the reviewer
> has found our work *relevant and worthy of discussion*, and is inclined for acceptance.
> Below, we have prepared detailed responses.
>
> **W1. Additional Case-Studies**
> > [A1] We highly agree that incorporating previous lock-in case studies would benefit our work and motivate new solutions. As such, we have incorporated two case studies in **Appendix A: Lock-In Case Studies**, where we illustrate (i) the aircraft
> automation dependency that led to a fatal plane crash - which eventually lead to regulations requiring pilots to log mandatory manual flight hours without autopilot assistance, and (ii) a large-scale data breach at a monopolistic e-commerce platform,
> where consumers continued to use the service despite the privacy risks - which lead to fines and sanctions, yet failed to reduce user dependency due to the absence of alternatives.
>
> **W2. Counterargument on Alternatives**
>
> >[A2] Yes, this is not just old wine in new bottle for two main reasons. Due to limited space, we kindly refer the reviewer to our reply in Reviewer 4 (uiGD)-[A1] for detailed reasons. Briefly, (i) Deskilling has shifted from physical execution to cognitive tasks, making AI-induced deskilling more consequential, affecting human cognitive skills (ii) AI-induced deskilling is self-undermining in a way that prior automation was not. For instance, the progressive offloading of cognitive tasks to AI is reducing the capability and credibility of human oversight, which many AI-safety work assumes for granted. Based on this, we have added a detailed counterargument to the **Alternatives** section.
>
> **[W3, Q1] Role of AI Researchers**
> > [A3] We thank the reviewer for this suggestion. Due to the word limit, we kindly request the reviewer to check our response to Reviewer 1(WFb4)-**W2-B. How the AI Research Community Can Contribute.** We also note that we have added **Appendix C- "Role of AI Researchers"**.
>
> **Q2. Open-Source Risk**
> > [A4] Here, we respectfully clarify a subtle but important distinction in our argument. Our claim in Section 4.1 is not that the adoption of open-source software leads to AI Lock-In, but rather that as AI Lock-In deepens, the pre-existing interdependencies within open-source ecosystems become increasingly dangerous.
>
> >Regarding the comparison to existing critical infrastructure failures (e.g., Linux), we believe that the risk under AI Lock-In would be more devastating. Failures in the current infrastructure cause functional disruption, where engineers have the capability to reason and make decisions. However, under serious AI Lock-In, where everything has been automated (delegating our decisions), we may lack the ability to make informed decisions.
>
> **Q3. Examples of Cyber Attack**
> > [A5] A recent example is the supply chain attack targeting LiteLLM, a popular Python open-source package. Adversaries injected malicious code into specific releases, leading to leakages of API (SSH) keys and credentials. Crucially, any package that uses LiteLLM as a dependency may automatically pull in the compromised version upon update. We have updated our paper to incorporate this example in **Sec 4.2**.
>
> **Q4. Vendor Lock-In**
> > [A6] We view this analogy as valid and believe it could strengthen the case for addressing vendor
> lock-in. However, as noted in our paper, vendor lock-in and AI lock-in operate at different levels, and our position was
> to focus on the more fundamental risk (i.e., AI Lock-In). We kindly refer the reviewer to our response to Reviewer 1(WFb4)-[W1], where we address this distinction in further detail.
>
> **Q5. Closed-Nature of LLM data**
>
> > [A7] We agree that the closed nature of commercial LLMs limits direct verification of biases. However, we note that the questions in Table 2 are not designed to yield definitive answers, but to cultivate **critical thinking** so that users do not passively accept AI outputs as truth. As such, users should treat AI outputs with the awareness that they may reflect biases embedded in training data that is itself undisclosed, making careful scrutiny all the more essential. We further acknowledge that this closedness is itself a structural problem that the AI research community, including at ICML, should address through research.
>
> **Q6. Building Robust Infrastructure as Risk Management
> > [A8] We acknowledge that robust infrastructure can reduce the risk of accidental AI failures. However, this does not address the fundamental risk of AI Lock-In: individuals lack the ability to critically reason and make informed decisions. Moreover, just because AI can be accessed without any interruption does not fundamentally deal with AI's hallucination and unreliable decisions. Our core position is that in an AI-locked-in society, the ability to identify and question such problems will be lost.
>
> We thank the reviewer for the detailed and considerate review. We remain available for further discussion if any other questions remain.

---

> > ### Author Rebuttal · Reviewer_gPWu · 2026-04-02
> >
> > Thanks for the detailed response! I think these additions resolve my concerns substantively, and will raise my score accordingly. There could probably be slightly better initial framing in the paper to avoid the sort of "how is this different from other lock-in types" question that the reviewers seem to have had, but I think these additions should address the question suitably if it arises for the reader.

---

### Official Review · Reviewer_WFb4 · 2026-03-15

**Significance:** 3
**Argument Clarity:** 3
**Rating:** 6
**Confidence:** 4

**Questions:**

See points under weaknesses.

**Alternative Views Section:**

Yes

**Compliance With Llm Reviewing Policy A Conservative:**

Affirmed.

**Discussion Potential:**

3

**Final Justification:**

It was a strong submission from the outset with a clear position that would warrant good discussions. My main concerns were adequately addressed by them in the rebuttal.

**Paper Summary:**

Use of AI by individuals and organizations without considering the long-term implications, particularly the dependency on these tools is discussed in this work. The authors hypothesise that this dependency will lead to AI Lock-In, wherein entities are so reliant on AI that this could lead to deskilling, and loss of skills/capabilities without the tools. The work uses lock-in theory from economics to formulate AI Lock-In, and identify different scenarios where AI Lock-In can have adverse impact. The paper concludes with three points under the call-to-action which can mitigate the effects of AI lock-in which focus on education and resilience building.

**Position:**

Yes

**Position In Title:**

Yes

**Related Work:**

3

**Strengths And Weaknesses:**

### Strengths

* The issue of AI Lock-in identified and addressed in this work is timely and the arguments provided are well-motivated. The use of Lock-In theory (Arthur 1989) and bet-hedging (Serger & Brockman, 1987) are appropriately used to formalise the risks of AI lock-in.

* Empirical evidence using several current examples highlight the criticality of over-dependence on AI. For example, the stackoverflow questions and ChatGPT traffic makes a good point about appropriation of common knowledge into AI ecosystems that can lead to deskilling and/or centralisation of knowledge.

* The use of koalas analogy is a nice addition to illustrate the risks of lock-in.

* The call to action points are reasonable. In particular, the point about AI-independent literacy is an important one.

### Weaknesses

1. Why do the authors refrain from extending the AI lock-in to also include vendor lock-in that is also ongoing due to the monopolistic nature of AI companies? They mention this in relation to sovereign AI, however, AI vendor lock-in poses compounded challenges when trying to overcome the dependence on AI.

2. The adaptation of media literacy to AI literacy in Table 2 is interesting. How can this be operationlised? In its current formulation, it reads like a group of slogans. Are there tools and infrastructure present to implement these recommendations? What is missing, and how can the AI research community like at ICML contribute towards these goals? This is a general comment about the entire paper that it could focus the arguments to better engage the research community, wherever possible.

3. How was the data for Fig. 1 obtained? Is it from Zhong & Wang (2024) cited in the caption. This must be clarified.

4. I think the title and the stated position in the paper needs to be elaborated some more and be nuanced than stating "we must be prepared". Perhaps interleaving some of the call-to-action points can be useful. For instance, "...., and we must build AI resilience with AI-Engaged and AI-Independent literacy", or something along these lines.

**Support:**

3

---

> ### Author Rebuttal · Authors · 2026-03-31
>
> We sincerely thank the reviewer for the detailed evaluation and for providing constructive feedback.
> We are happy to see that reviewer WFb4 has found our work *timely and well-motivated*, and pointing
> out that *Our proposal is an important one.* In the below we have prepared detailed responses.
>
> **W1. Why Not Extend to Vendor Lock-In?**
>
> > [A1] We thank the reviewer for this important point. We clarify that Vendor Lock-In is about being locked to a specific AI provider, while AI Lock-In is about being overly dependent on AI itself. Briefly, solving Vendor Lock-In does not solve AI Lock-In, but solving AI Lock-In can solve Vendor Lock-In on AI services. As such, **AI Lock-In is the more fundamental problem**, while Vendor Lock-In is a separate risk that, when it overlaps with AI-Lock In, it can have a compounding effect. We believe that both are important problems, but we intentionally focused on the more root challenge to maintain clarity. However, based on the reviewer's feedback,
> we found that the paper would benefit from delivering such a distinction clearly.
> As such, we have added a subsection **Distinguishing AI Lock-In from Vendor Lock-In** under the Discussion section to emphasize our focus.
>
> **W2-A. How Can We Operationalize Table 2 in Scale? Also, What is Missing?**
>
> > [A2] We thank the reviewer for this feedback and acknowledge that Table 2 remains at a conceptual level. We propose packaging the five concepts into a structured AI literacy workshop for schools, organizations, and national programs.
> Each concept translates into a concrete exercise:
>
> > - (1) Authorship → source verification habits, training users to request and verify original sources for AI claims;
> > - (2) Format → confidence calibration exercises where learners encounter AI providing confidently worded but incorrect answers;
> > - (3) Audience → filter bubble comparisons where workshop participants pose the same question to AI and compare each other's results, directly experiencing how individual prompting habits shape AI's filter bubble;
> > - (4) Content → perspective gap analyses where learners examine which cultural viewpoints AI includes or omits (e.g., asking AI to "describe a typical breakfast" yields only Western dishes like pancakes and bacon, revealing biases in training data);
> > - (5) Purpose → engagement design awareness, analyzing how AI services sustain usage (e.g., ending responses with questions to maintain conversations).
>
> >**What Is Missing.** Implementing this workshop at scale requires infrastructure that does not yet exist.
> Most critically, there is no standardized instrument to track how AI dependency changes over time.
> Without such measurement, assessing whether AI Lock-In is deepening or interventions are effective remains difficult.
>
> > We have added **Appendix B: Operationalizing AI Literacy in Scale based on Media Literacy** for a more detailed discussion.
>
> **W2-B. How the AI Research Community Can Contribute**
>
> > [A3] We propose developing the following for the AI research community.
>
> >**(i) Measurement:** We need to understand the current state of AI Lock-In. Such an objective can be achieved by developing an AI Lock-In Index for international and national organizations to periodically monitor AI dependency. Development of benchmarks that can assess deskilling is also recommended.
>
> >**(ii) Designing:** We need to consider AI Lock-In as an important alignment factor that needs to be considered in model designing. Many alignment works focus on user retention, safety, but often dismiss "how much we are preserving human agency". Such technical research directions need to be supported by organizations and nations.
>
> >**(iii) Testing:** We need evaluation frameworks that can assess how much we or as organizations are dependent on AI. Similar to red teaming research in adversarial robustness, a similar approach can be considered.
>
> >Based on the above, we have added **"Appendix C: How the AI Research Community can Directly Contribute"** in the updated paper.
>
> **W3. Source of Fig. 1**
>
> > [A4] Thank you for the opportunity to clarify our source.
> StackOverflow statistics were obtained from the StackExchange Data Explorer, and ChatGPT traffic estimates were sourced from SimilarWeb via DemandSage. We have updated the manuscript to clarify the source.
>
> **W4. Title of paper**
>
> >[A5] We value the reviewer's thoughtful suggestion.
> Our intention with the current title, “AI Lock-In Is in Progress, and We Must Be Prepared,”
> was to deliberately keep the framing broad and accessible, emphasizing the urgency and generality of the problem.
> As a position paper, we aimed for the title to work as an entry point that highlights the
> existence and importance of the risk. We are also concerned that emphasizing a specific call-to-action in the title might give the impression that the solutions are already fully established, when in fact, AI Lock-In is an ongoing challenge that requires continued research from the community.

---

> > ### Author Rebuttal · Reviewer_WFb4 · 2026-04-02
> >
> > The authors have addressed my main concerns sufficiently. I would still suggest adjusting the position from its current form. However, I would also be okay with the current version.
> >
> > I will raise my score.

---

### Decision · Program_Chairs · 2026-04-30

**Decision:**

Accept (spotlight)

**Comment:**

The reviewers broadly agree that the “AI lock-in” framing is valuable, timely, and useful for stimulating discussion. The paper’s main strengths is 1) its clear framing of AI-dependency as a neglected safety risk, supported by individual-level evidence for cognitive offloading and de-skilling, 2) the use of lock-in theory to provide a solid conceptual framework for the analysis, 3) the distinction between AI-engaged and AI-independent literacy, and 4) the broader claim that safety research should not only consider harmful outputs from AI, but also harmful dependence on AI. The main argument becomes more speculative as it moves from assessing individual harms to organizational/national risk, and hte empirical support is somewhat uneven and correlational. Additionally, the paper initially underdeveloped some key comparisons (e.g, to prior automation, infrastructure/tool/software/vendor lock-in, and the related human-factors literature. The rebuttal appears to have addressed many of these concerns by improving the paper’s novelty claims, adding case studies and related work, and clarifying other aspects raised in the initial reviews. Overall, it remains a compelling and thought-provoking paper.